# Rational Design of Lipid-Based Vectors for Advanced Therapeutic Vaccines

**DOI:** 10.3390/vaccines12060603

**Published:** 2024-05-31

**Authors:** Yufei Ma, Yiang Chen, Zilu Li, Yu Zhao

**Affiliations:** 1Meinig School of Biomedical Engineering, Cornell University, Ithaca, NY 14853, USA; ym538@cornell.edu; 2College of Chemistry, Nankai University, Tianjin 300071, China; 2120200822@mail.nankai.edu.cn

**Keywords:** lipid-based vectors, synthetic lipids, natural lipids, lymphoid organ-targeting, endosomal escape

## Abstract

Recent advancements in vaccine delivery systems have seen the utilization of various materials, including lipids, polymers, peptides, metals, and inorganic substances, for constructing non-viral vectors. Among these, lipid-based nanoparticles, composed of natural, synthetic, or physiological lipid/phospholipid materials, offer significant advantages such as biocompatibility, biodegradability, and safety, making them ideal for vaccine delivery. These lipid-based vectors can protect encapsulated antigens and/or mRNA from degradation, precisely tune chemical and physical properties to mimic viruses, facilitate targeted delivery to specific immune cells, and enable efficient endosomal escape for robust immune activation. Notably, lipid-based vaccines, exemplified by those developed by BioNTech/Pfizer and Moderna against COVID-19, have gained approval for human use. This review highlights rational design strategies for vaccine delivery, emphasizing lymphoid organ targeting and effective endosomal escape. It also discusses the importance of rational formulation design and structure–activity relationships, along with reviewing components and potential applications of lipid-based vectors. Additionally, it addresses current challenges and future prospects in translating lipid-based vaccine therapies for cancer and infectious diseases into clinical practice.

## 1. Introduction

Vaccination stands as one of the most potent medical interventions ever devised by humanity, heralding unparalleled reductions in medical expenses, human anguish, and mortality rates. Traditional vaccines, relying on live attenuated pathogens, are renowned for their ability to trigger both humoral and cellular immunity [1,2]. However, their safety is compromised by the potential for the attenuated pathogen to revert to a pathogenic state, posing a risk of infection. Subunit vaccines offer a safer alternative but are less effective in eliciting the cellular immunity crucial for combating intracellular pathogens. Emerging vaccine technology utilizing viral vectors shows promise for inducing robust humoral and cytotoxic T cell responses, as well as antigen-specific immunological memory [2,3]. Nevertheless, concerns regarding their high immunogenicity impede the full realization of their clinical potential as delivery systems [4]. In contrast to viral vectors, non-viral vectors exhibit lower immunogenicity. The non-viral nanocarriers serve as stable vectors capable of efficiently delivering antigens or messenger RNA (mRNA) expressing targeted antigens into antigen-presenting cells (APCs) within lymphoid organs, consequently eliciting robust immune responses in vivo [5]. Moreover, the efficacy of this type of vaccine can be improved by optimizing delivery formulations.

In recent years, various materials, including lipids, polymers, peptides, metals, and inorganic materials, have been employed for the construction of non-viral vectors, and their properties can be finely tuned as needed [6]. But the risks and benefits must be carefully balanced before design and preparation. Among these vectors, lipid-based nanoparticles, typically ranging from 100 to 1000 nm in diameter, offer the advantages of biocompatibility, biodegradability, and safety, making them an ideal choice for vaccine delivery systems [7]. Specifically, (a) lipid-based vectors can effectively protect the encapsulated antigens or mRNA from degradation by proteases or ribonucleases and self-hydrolysis, as well as aggregation in the body [8]; (b) the chemical and physical properties (e.g., material composition, geometry, surface chemistry, and charge) of lipid-based vectors can be precisely tuned to mimic viruses, enabling efficient draining through the lymphoid system and internalization by APCs within lymphoid organs (e.g., lymph nodes (LNs) and spleen); (c) surface modification of lipid-based vectors offers additional feasibility to further improve targeted delivery to specific immune cells, through conjugating the receptor ligands or antibodies; (d) lipid-based vectors can be engineered to facilitate endosomal escape, thereby improving the cytosolic delivery of antigens or mRNA within APCs, which enables robust antigen cross-presentation and cytotoxic T cell activation; (e) antigens or mRNA and adjuvants can be conveniently co-delivered into the same APCs using lipid-based vectors, for optimal immune activation; and (f) lipid-based vectors formulated with biocompatible materials characterized by low immunogenicity can overcome anti-vector immunity, which allows for multiple injections of the same formulation without loss of potency, presenting a significant advantage over viral vectors. Because of the above advantages, two lipid-based vaccines developed by BioNTech/Pfizer (BNT162b2) and Moderna (mRNA-1273) became the first vaccines to receive conditional approval for use in humans against coronavirus disease 2019 (COVID-19) [9].

Various reviews on lipid-based nanoparticles have been published in recent years, and this review aims to provide an updated introduction to the advances and challenges in rational design of advanced vaccine delivery systems. In this review, we highlight recent rational design strategies for the purpose of delivery of vaccines to the lymphoid organs and effective endosome escape (Figure 1). Although high-throughput screening methods can obtain optimal formulations, the rational design of formulations remains a crucial aspect that cannot be overlooked. This work also reviews the components (including synthetic and natural lipids) used to construct lipid-based vectors for vaccine therapies and outlines their respective advantages and potential applications (Table 1). In addition, the current challenges and future development in lipid-based vectors for the clinical translation of vaccine therapies against cancer and infectious diseases will be highlighted.

## 2. Lymphoid Organ-Targeting and Endosomal Escape

Although lipid-based vaccines have been clinically used to combat COVID-19, certain immunological elements still require refinement to enhance the outcomes. For example, the durability of the antibody response against targeted antigens is short-lived, requiring the administration of booster doses to maintain the protection against infection. In cancer vaccine therapy, cancer vaccines represent a much higher bar than COVID-19 vaccines [10]. This is because to effectively activate the body’s immune system, tumor antigens need to be efficiently cross-presented in APCs, which involves multiple steps, mainly involving the accumulation of vaccines in lymphoid organs via homing effects and endosomal escape in APCs [11]. Lymphoid organs contain large numbers of lymphocytes and play an important role in the initiation of robust and durable immune responses [12]. Indeed, the fate of vaccines is not solely confined to APCs with lymphoid organs. Following intramuscular (i.m.) administration, the majority are internalized by local muscle cells, while a residual portion enters the bloodstream and accumulates in the liver, resulting in the emergence of off-target effects and side effects [13]. Moreover, vaccines typically gain access to the interior of APCs through endocytosis pathways. Most of the vaccine that enters the cells is entrapped within endosomes, resulting in only a small amount (1~2%) of vaccine entering the cytoplasm to function [14]. In other words, the accumulation efficiency of a vaccine in the lymphoid organs (LE) and the efficiency of endosome escape (EE) collectively affect the ultimate therapeutic outcomes. In this section, we focus on how to rationally design and optimize lipid-based vectors to effectively deliver vaccines into the lymphoid organs and facilitate cellular uptake and subsequent endosome escape.

By adjusting the physical and chemical properties of lipid-based vaccines, they can be directed towards the lymphoid organs. For instance, altering their surface charge and/or particle size has been demonstrated to modify their biodistribution. Siegwart et al. developed a charge-based selective organ targeting (SORT) lipid nanoparticle (LNP) platform for organ-specific mRNA delivery [15,16,17]. They discovered that negatively charged lipids directed mRNA transfection toward the spleen. Subsequent mechanistic studies revealed that the protein corona formed on the surface of the nanoparticles plays a crucial role. More recently, Wang et al. also found that noncationic lipid-based vectors showed spleen targeting delivery ability [18]. The size of the vector also determines its fate in the body. For example, after intradermal (i.d.) injection, interstitial flow transported ultra-small nanoparticles (25 nm) highly efficiently into lymphatic capillaries and their draining LNs, targeting half of the LN-residing APCs, whereas 100 nm nanoparticles were only 10% as efficient [19]. In addition, integrating ligands or antibodies with cell-targeting functions into lipid molecules can also enhance the lymphoid-organ-targeting capability of the vectors [20].

Lipid-based vectors with increased endosomal escape capabilities are optimal for vaccine delivery. These vectors typically enter the APCs via endocytosis pathway, which traps most internalized vectors (>98%) in the endosomes characterized by a pH value of 5.5 to 6.3, which subsequently evolve into lysosomes with numerous digestive enzymes and a pH value of 4.5 to 5.5. This results in degradation of vaccines and the failure of the vaccines to reach the cytoplasm to function. This means that the pH difference between the inside and outside of cells can serve as a trigger to promote endosomal escape. The mechanism of endosomal escape involves destabilization of the endosomal membrane by increased interaction between the endosomal membrane in a pH-sensitive manner [21]. Current strategies used by lipid-based vectors for endosomal escape include: (a) “proton sponge effect” mediated by a high pH-buffering material that swells when protonated; (b) pH-responsive charge switching that increases the interaction with endosomal membranes; and (c) membrane fusion-mediated endosomal escape by integrating fusogenic lipids into vectors. Recent studies found that vector-topology (cuboplex nanostructure) mediated membrane fusion and that mechanical-action-induced endosomal membrane damage can also improve the endosomal escape efficiency [22,23].

The “proton sponge effect” facilitates endosomal escape by increasing osmotic pressure within endosomes, leading to swelling and rupture. This process, driven by pH-buffering materials, enables a simultaneous influx of protons and chloride ions, disrupting endosomes and allowing vaccine escape. For example, Liu et al. integrated 2-(hexamethyleneimino)ethyl groups into a lipid-based vector to enhance its endosomal escape ability [24,25]. As a result, after entering the APCs, the vectors facilitated the escape of the loaded tumor antigen from endosomes due to the protonation of 2-(hexamethyleneimino)ethyl groups and subsequent antigen cross-presentation, leading to enhanced priming of cytotoxic T cells. Commercially available LNPs also show the ability to protonate, but mechanistic studies have shown that they seem to prefer to interact with the endosomal membranes directly, thereby inducing destabilization [26]. Specifically, following cellular uptake, the ionizable lipids in LNPs undergo protonation within acidic endosomes and interact with anionic endosomal phospholipids through electrostatic interactions, forming cone-shaped ion pairs. These cationic–anionic lipid pairs induce a transition from bilayer to inverted hexagonal H_II_ phase, resulting in membrane disruption for effective endosomal escape. Enhancing the fusion ability of lipid-based vectors with an endosomal membrane also shows the potential to improve escape efficiency. Integration of 1,2-dioleoyl-sn-glycero-3-phosphoethanolamine (DOPE) and 1-palmitoyl-2-oleoyl-glycero-3-phosphocholine (POPC) into lipid-based vectors has been shown to introduce membrane curvature and increase tension, which in turn facilitates membrane fusion [27,28,29]. Xu et al. found that DOPC could significantly facilitate the mRNA transfection at both cell and organ levels, compared with 1,2-distearoyl-sn-glycero-3-phosphocholine (DSPC) with limited fusion efficacy [30]. Recently, Gu et al. reported that fluorinated modification of the 1,2-distearoyl-*sn*-glycero-3-phosphoethanolamine-poly(ethylene glycol)-2000 (PEG-DSPE) could improve mRNA transfection [31]. A potential mechanism is that fluorinated lipids are more likely to fuse with the endosomal membrane, thereby promoting escape.

## 3. Lipid-Based Vectors for Constructing Therapeutic Vaccines

### 3.1. Synthetic Lipid-Based Vectors

#### 3.1.1. Lipid Nanoparticles

LNPs have gained significant interest in recent years due to their efficacy as non-viral carriers for nucleic acid delivery [32]. These nanoparticles are characterized by a single layer of lipids, consisting of ionizable lipids, helper phospholipids, cholesterol, and polyethylene glycolylated (PEGylated) lipids, which together constitute the outer shell [33]. With the recent clearance by the FDA of two vaccines based on LNPs for COVID-19, there is a growing interest in exploring innovative LNP-based therapies. In this section, we summarize the applications of LNPs for combatting other infectious disorders and cancers.

Recent innovations in LNP technology have shown promising results in enhancing the efficacy of in situ cancer vaccinations and targeted therapeutic delivery. In a recent study, Xu et al. explored a LNP design that significantly enhanced in situ cancer vaccination [34]. By incorporating 93-O17S-F/cGAMP, this approach boosted cross-presentation of tumor antigens and activated the STING pathway, leading to a notable anti-tumor efficacy in mice. Remarkably, this strategy not only eradicated primary tumors in 35% of the cases but also fostered immune memory, enabling 71% of the mice to fend off tumor rechallenge (Figure 2a). Another study by the same group explored LNP-based LN-targeting delivery systems, demonstrating improved cytotoxic CD8^+^ T cell responses and enhanced protective effects against melanoma, indicative of the system’s potential as a platform for future mRNA cancer vaccines. These innovations underline that strategic design enhancements in LNPs can directly impact therapeutic efficacy and immune response in cancer treatments. Al-Wassiti et al. demonstrated that mRNA vaccines, delivered via LNPs to secondary lymphoid organs after intramuscular injection, are critical for eliciting strong adaptive immune responses in mice [35]. They compared responses to LNPs carrying luciferase and ovalbumin (OVA) mRNAs, revealing that direct mRNA delivery to these organs, rather than local muscle effects, significantly enhanced immune reactions. The adaptation of LNPs for mRNA delivery has revolutionized the development of next-generation therapeutic agents, highlighting the importance of biomimetic strategies. Jiang et al. added phosphatidylserine (PS) to the standard four-component MC3-based LNP formulation (PS-LNP), for efficient mRNA delivery to secondary lymphoid organs via systemic administration, utilizing a biomimetic strategy to enhance cellular uptake by immune cells [20]. This was attributed to the fact that PS is a well-known signaling molecule that promotes the endocytic activity of phagocytes, including monocytes and macrophages. Beyond cancer vaccines, the role of LNPs in targeting respiratory pathogens through mRNA vaccines is exemplified by their strategic design for enhancing vaccine efficacy, particularly for diseases affecting the respiratory system (Figure 2b). Doroudian et al. illustrated how LNPs effectively deliver mRNA vaccines against respiratory viruses [36]. They highlighted the crucial LNP components—cationic and ionizable lipids, PEGs, and cholesterol—that facilitate mRNA’s cytoplasmic transfer, thereby enhancing vaccine effectiveness and stability.

Endosomal escape is pivotal for intracellular delivery of vaccines, which involves transporting vaccines directly into cells, to target specific cellular processes and enhance the effectiveness of treatments at the molecular level. Xu et al. introduced an innovative mechanism via lipid-based nanoscale molecular machines, which utilized a light-induced nanomechanical action to destabilize endo-lysosomal compartments, enhancing the cytosolic delivery of therapeutic agents [23]. Building on this concept of improving delivery efficacy, Choi et al. further advanced LNP technology by incorporating histidinamide-conjugated cholesterol into LNP formulations, thereby improving mRNA delivery by facilitating endosomal escape [37]. This modification led to potent immune responses and increased stability of the mRNA within biological systems. These advancements highlight the significant progress in overcoming the barriers to efficient intracellular delivery of therapeutic molecules.

The rational design of lipid-based vectors has become crucial in the development of advanced therapeutic vaccines targeting cancer. Mitchell et al. designed an adjuvant lipidoid that could be partially substituted into LNPs to improve the adjuvanticity of mRNA vaccines [38]. This innovative design not only enhanced mRNA delivery but also endowed the LNPs with Toll-like receptor 7/8-agonistic activity, resulting in a significant boost in the innate immune response, thereby illustrating LNPs’ capability to enhance protective immunity against viral infections (Figure 2c). Similarly, Dong et al. synthesized and evaluated a series of non-nucleotide STING agonist-derived amino lipids, which were incorporated into LNPs [39]. These LNPs, particularly SAL12-LNPs, proved most effective in delivering mRNA encoding SARS-CoV-2 Spike protein and activating the STING pathway in dendritic cells, further showcasing LNPs’ capability to enhance protective immunity against viral infections. Zhou et al. emphasized that the surplus positive charge of lipids is essential for condensing mRNA via electrostatic interaction, but it leads to adverse reactions such as inflammation and cytotoxicity [33]. Recently, Deng et al. developed noncationic thiourea LNPs (NC-TNPs) that compact mRNA through robust hydrogen bonding between the thiourea groups in NC-TNP and the phosphate groups of mRNA, still resulting in a high accumulation ratio (spleen/liver) [18]. Importantly, this formulation also significantly reduced the inflammation and toxicity at the injection site. 

**Figure 2 vaccines-12-00603-f002:**
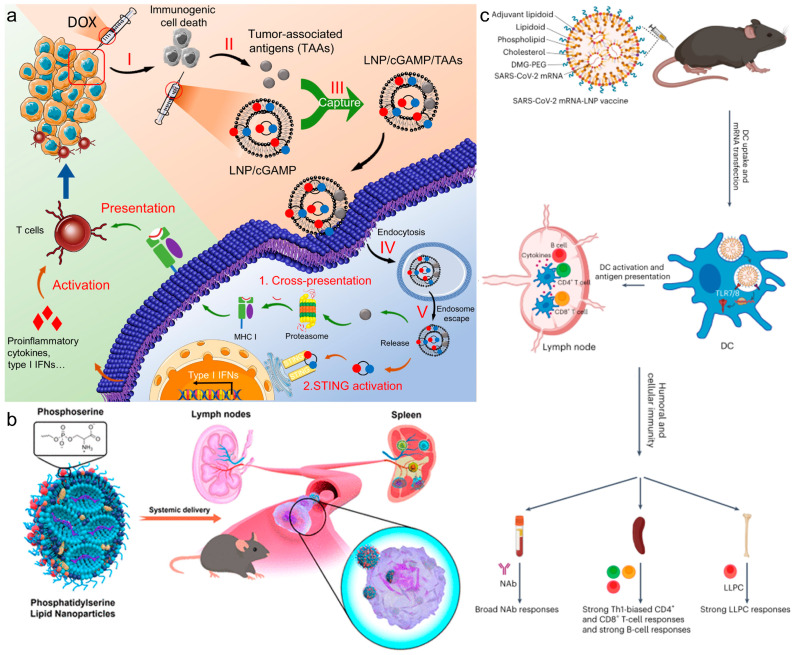
(**a**) In situ cancer vaccination using lipidoid nanoparticles. Reproduced from ref. [34]. Copyright 2021 AAAS. (**b**) Phosphatidylserine lipid nanoparticles promote systemic RNA delivery to secondary lymphoid organs. Reproduced from ref. [20]. Copyright 2022 American Chemical Society. (**c**) Adjuvant lipidoid-substituted lipid nanoparticles augment the immunogenicity of SARS-CoV-2 mRNA vaccines. Reproduced from ref. [38]. Copyright 2023 Nature Publishing Group.

LNPs can also be used for ex vivo engineering of immunosuppressive T cells for autoimmunity therapies. Regulatory T (T_reg_) cell is an immunosuppressive cell characterized by its expression of the transcription factor Foxp3. It has been reported that engineering Foxp3 expression in CD4^+^ T cells can result in a T_reg_-like phenotype. Mitchell et al. developed a LNP platform to deliver Foxp3 mRNA to CD4^+^ T cells [40]. They successfully engineered CD4^+^ T cells into Foxp3-T (FP3T) cells that transiently exhibited an immunosuppressive phenotype and functionally suppressed the proliferation of effector T cells. These findings highlight the potential of LNP platforms for modifying T cells to exhibit immunosuppressive properties, offering prospects for the development of therapies targeting autoimmune conditions. Most recently, Xia et al. reviewed antigen-specific mRNA LNP platforms for the prevention and treatment of allergy and autoimmune diseases [41].

#### 3.1.2. Liposomes

Liposomes, a prevalent type of lipid-based vector, are integral in the development of therapeutic vaccines, due to their unique structure and versatile functionality. Similarly to LNPs, liposomes are composed of a lipid bilayer that can encapsulate various drugs due to its hydrophobic region and a hydrophilic core. Distinctly from LNPs, liposomes are categorized by the number of lipid bilayers they contain. Unilamellar liposomes consist of a single lipid bilayer, and are typically utilized for their simplicity and efficiency in drug delivery, whereas multilamellar liposomes feature multiple layers that can enhance drug load and enable controlled release [42]. These liposomes vary in size from 25 nm to 2.5 µm, with both size and surface charge playing crucial roles in their biodistribution and cellular interactions [43,44,45]. In vaccine therapy, liposomes have been effectively used to deliver both antigens and adjuvants directly to DCs, promoting maturation and initiating strong anti-tumor responses [46]. By manipulating their size, charge, and bilayer structure, liposomes offer a robust platform for novel vaccine strategies, significantly impacting various stages of the cancer immunity cycle. This includes sophisticated approaches that combine multiple therapeutic mechanisms to enhance the efficacy of treatments like immune checkpoint inhibitors, illustrating their potential to profoundly influence advanced therapeutic vaccines [47,48].

Liposomes excel in enhancing targeted delivery to specific immune cells. They encapsulate drugs and can be modified to bind selectively to desired cells, focusing treatment and improving outcomes. Wang et al. reviewed the importance of modifying liposomes with immunopotentiators and targeting molecules, improving their capacity to induce strong immune responses [49]. For example, Haan et al. focused on targeting CD169/Siglec-1^+^ APCs with liposomes, effectively enhancing cytokine production and T cell activation [50]. This specificity in targeting was further explored by Chen’s group, who used synthetic liposomal constructs to mimic biological viruses, providing a novel tool for studying B-cell responses and immune activation [51]. Similarly, the research of Hussein’s group on poly(ethylenimine)-based liposomes highlighted their enhanced immune response against bacterial infections like Group A Streptococcus, demonstrating the potential of targeted liposome delivery [52].

The ability to control the release of encapsulated agents is another crucial advantage of liposomes. For example, the cobalt-porphyrin-phospholipid (CoPoP) liposome system developed by Lovell et al. demonstrated how liposomes can efficiently sustain the release of CD8^+^ T cell epitopes, enhancing the cellular immune responses essential for effective cancer vaccines [53]. The work from the same group with lyophilized, thermostable liposomes also underscored their ability to maintain stability and induce protective immunity under various storage conditions, which is crucial for vaccines like those for SARS-CoV-2 [54]. In addition, combining different therapeutic mechanisms within liposomal formulations can lead to more effective treatments. Chen et al. integrated checkpoint blockade with nanosonosensitizer-augmented sonodynamic therapy using liposomes, reducing tumor growth and metastasis effectively [55]. Kent et al. enhanced germinal center and follicular helper T cell responses using hemagglutinin-functionalized liposomes, showing potential against viral pathogens like influenza [56].

#### 3.1.3. Lipid-Based Hybrid Nanoparticles

Polymers, sourced naturally (e.g., chitosan, alginate) or synthetically (e.g., poly(lactic-co-glycolic acid) (PLGA) and poly(caprolactone) (PCL)), are pivotal biomaterials in biomedicine. Natural types offer superior biocompatibility, while synthetic forms allow tailored properties. Like lipids, polymers serve as drug carriers, offering stability, high drug loading, and adjustable degradation. Hybrid lipid–polymer nanoparticles combine these advantages, promising innovative delivery systems.

One of the most common approaches for in situ vaccination is to enhance immunogenic cell death (ICD) of tumor cells to elicit an anti-tumor response. Yang et al. engineered a hybrid nanoparticle with a PLGA polymeric core and an outer lipid shell, encapsulating both the chemotherapeutic agent DOX and Ce6, a photosensitizer for photodynamic therapy (PDT). The polymeric core degradation by ROS facilitated a cascading effect, triggering the release of the chemotherapeutic agent post-PDT treatment. This approach augmented tumor ICD through multiple mechanisms, thereby enhancing the in vivo anti-tumor response and leading to reduced tumor growth [57]. In general, in situ vaccination strategies, such as this one, are most effective for tumors with high mutational burdens, and they offer promise for addressing tumor heterogeneity. By employing lipid–polymer hybrid nanoparticles, in situ vaccination can be combined with other immunotherapeutic approaches to provoke a potent anti-tumor response. Recently, Stephenson et al. used polyethylenimine (PEI)-coated liposomes to enhance both mucosal and systemic immunity, providing a strong defense against pathogens [58]. While the polymeric core enveloped by a lipid shell is a prevalent configuration for lipid–polymer hybrid nanoparticles, innovative designs are emerging for novel immunotherapeutic strategies. Recent studies by Liu et al. showcased an anchored external polymeric network, enhancing the stability and endosomal escape capabilities [24,25]. This two-pronged approach enhanced tumor ICD via PDT sensitizers within the hybrid particle, while the maleimide groups on its surface captured tumor-associated antigens, facilitating their delivery to DCs for maturation. This hybrid particle markedly enhanced DC and CD8^+^ T cell activation, extending the survival of 4T1 tumor-bearing mice to 60 days (Figure 3a). Wang et al. presented an implantable blood clot loaded with liposomes-protamine-hyaluronic acid nanoparticles (LPH NPs) carrying both a vaccine (LPH-vaccine) and siRNA (LPH-siRNA), for synergistic DC recruitment and activation [59]. The subcutaneously implanted blood clot scaffold efficiently attracted numerous immune cells, particularly DCs, fostering a DC-rich environment in vivo, suggesting a promising strategy for cancer therapy in the clinic.

Loading both antigen and adjuvant onto a single particle enhances DC maturation and subsequent T cell priming [60]. Zhu et al. demonstrated this using a lipid–polymer particle to deliver both antigen and two Toll-like receptor (TLR) agonists [61]. One agonist, IMQ, was loaded into the polymeric core to activate TLR 7 in the endo/lysosome upon cellular uptake, while the monophosphoryl lipid A (MPLA) in the outer lipid shell interacted with TLR 4 in the DC bilayers. Moreover, OVA was adsorbed onto the particle surface as a representative antigen. This strategic placement of TLR agonists synergistically enhanced DC maturation and induced Th1-skewed humoral and cellular immune responses against EG7-OVA tumors in vivo.

Incorporating inorganic components into lipid-based vectors imparts distinctive properties to lipid–inorganic hybrid nanoparticles, facilitating the development of innovative immunotherapeutic strategies. Common inorganic elements used include mesoporous silica, gold, and iron oxide nanoparticles. These hybrids retain key properties of lipid-based vectors, while gaining additional functionalities. For instance, in strategies employing iron oxide nanoparticles encapsulated in lipid-based vectors, the nanoparticle surface serves as a platform for conjugating adjuvants and antigens, ensuring their proximity. While the lipid coating extends the circulation time, the close association of adjuvant and antigen on the iron oxide nanoparticle surface enhances immune response upon uptake by APCs. Similarly, mesoporous silica nanoparticles offer beneficial porosity and facile surface functionalization, enabling unique co-delivery strategies.

Chertok et al. explored the biodistribution of hybrid lipid–inorganic nanoparticles featuring 10 nm or 30 nm iron oxide cores [62]. These hybrids not only facilitated enhanced adjuvant uptake in LNs but also enabled in vivo tracking via magnetic resonance imaging (MRI). T2-weighted MRI revealed areas with altered magnetic susceptibility, indicating the presence of hybrid particles in tissues. Although MRI is the primary modality for tracking such nanoparticles, they can be adapted for other imaging methods. For instance, Mareque-Rivas et al. radiolabeled iron oxide particle cores with Ga^+^ for nuclear imaging. Utilizing SPECT/PET imaging, they tracked inorganic-lipid particles for LN drainage [63]. In vivo particle tracking aids in investigating and tailoring particle biodistribution for optimal efficacy and therapeutic applications.

**Figure 3 vaccines-12-00603-f003:**
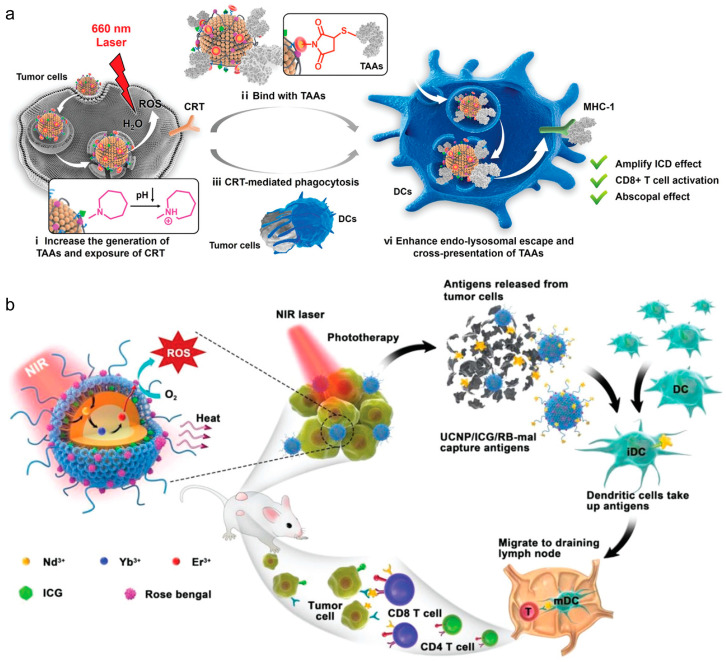
(**a**) Polymer-reinforced liposomes amplify immunogenic cell death-associated antitumor immunity for photodynamic-immunotherapy. Reproduced from ref. [24]. Copyright 2022 Wiley-VCH. (**b**) NIR-triggered phototherapy and immunotherapy via an antigen-capturing nanoplatform for metastatic cancer treatment. Reproduced from ref. [64]. Copyright 2019 Wiley-VCH.

Incorporating inorganic elements into lipid–inorganic hybrid particles has been shown to enhance PDT or photothermal (PTT) therapies for combating tumors. In PDT, these sensitizers generate reactive oxygen species (ROS) for ICD, while in PTT, they convert light into vibrational energy, causing thermal ICD. Chen et al. devised an in situ vaccination approach using upconverted nanoparticles coated with maleimide-functionalized lipids [64]. Upon irradiation, the Nd/Yb/Er core acted as a PDT sensitizer, inducing ROS and triggering tumor ICD, while lipid–maleimide structures captured tumor-associated antigens (Figure 3b). Similarly, Oh et al. demonstrated lipid–inorganic hybrid nanoparticles enhancing PTT by encapsulating gold nanoparticles [65]. Under near-infrared light, the gold nanoparticles acted as photosensitizers, heating tumors to ~50 °C, increasing tumor ICD, and transforming the immunologically cold tumor microenvironment into a hot one. These findings highlight that lipid–inorganic nanoparticles support immunotherapeutic strategies and enable unique combinatorial approaches for potent anti-tumor responses.

### 3.2. Natural Lipid-Based Vectors

Cell membranes, comprised of natural lipids, have gained significant traction in medical applications. In recent years, these membranes have emerged as popular coatings for lipid-based vectors. Essentially a lipid bilayer infused with proteins, the cell membrane represents a two-dimensional nanomaterial, approximately 10 nm thick, boasting diverse bio-functionalities. These novel vectors exhibit behaviors akin to their source cells, boasting enhanced biocompatibility, prolonged circulation time, and improved active targeting efficiency. Additionally, they offer an added layer of functionality mimicking cells, further enhancing their utility. In this section, cell membrane-coated nanoparticles and bacterial membrane-coated nanoparticles, as well as their potential applications, will be introduced.

#### 3.2.1. Cell Membrane-Coated Nanoparticles

##### Immune Cell Membrane

Immune cell membrane-coated nanoparticles (CNPs) derived from various immune cells, especially the macrophages and DCs, play a pivotal role in the development of innovative cancer treatments, particularly in vaccine delivery. These immune cells play a pivotal role in tumorigenesis, with the capacity to either hinder or facilitate cancer progression, depending on their environment and activation status [66]. Leveraging the innate characteristics of immune cells, including their tendency to accumulate at sites of inflammation—a frequent precursor to cancer—and their involvement in immunosuppression within tumor microenvironments, researchers have utilized their membranes to engineer CNPs. These CNPs demonstrate prolonged circulation in the bloodstream, precise targeting of tumors, and heightened immune responses against malignancies. CNPs hold significant promise for transforming the landscape of cancer immunotherapy and vaccine development. These innovative designs not only optimize therapeutic efficacy but also expand the possibilities for future medical applications in targeting various cancers and enhancing patient outcomes.

Macrophages and DCs have been studied for their unique interactions with tumors. They can migrate directly to tumor sites, facilitated by specific integrin interactions, making them ideal for delivering therapeutic agents directly to tumors [67]. Leveraging immune cell membranes, CNPs effectively target and deliver vaccines to enhance cancer immunotherapy. Utilizing these unique properties, CNPs can be engineered to enhance the delivery of vaccine components directly to tumor sites. For example, macrophage-based CNPs exploit the macrophages’ natural tumor-homing capabilities to deliver therapeutic agents such as siRNAs to suppress immune evasion pathways in tumors, as demonstrated in research on glioblastoma models. Similarly, DCs, known for their efficacy in antigen presentation, are used in CNPs to potentiate the immune system [68]. An important advantage of mature DC membranes is that they have a broad spectrum of peptide/MHC complexes on their surface, thus having the capability to present antigens that can activate T cells specifically. Therefore, in recent years, DC-based CNPs have become attractive application prospects for vaccine delivery. Liu et al. developed a DC-based nanovaccine by integrating SiPCCl_2_-hybridized mesoporous silica with Fe(III)-captopril to form the core, then enveloping it with DC membranes matured through H22-specific antigen stimulation [68] (Figure 4a). These nanovaccines effectively targeted H22 tumors, enhanced tumor-associated antigen release, and triggered ICD upon photodynamic treatment. The study demonstrated that these nanovaccines bolstered T cell-mediated anti-tumor responses against both primary and distal H22 tumors in mouse models, consequently extending their survival duration. Wang et al. developed lymphatic-targeting vectors using DC membranes, effectively reaching spleen and LNs. Nanoparticles with YK009 lipid ensured high mRNA encapsulation and endosomal escape. DC-based CNPs induced protein expression in spleen and LNs, generating Spike-specific IgG, neutralizing antibodies, and Th1-biased cellular immunity against SARS-CoV-2 [69] (Figure 4b). In another work, Zhang et al. reported DC-cancer fused membrane-based CNPs (NP@FMs) for targeted cancer immunotherapy. The fused membranes endowed NP@FMs with the capability to target homologous tumors and LNs. Therefore, NP@FMs triggered robust innate and adaptive anti-tumor immunity. Through combining PDT and immunotherapy, NP@FMs showed a strong ability to suppress distant tumor proliferation (Figure 4c) [70]. 

Immune cell CNPs also play important roles as vaccines in treating autoimmune diseases [71]. Rheumatoid arthritis (RA) is an autoimmune condition characterized by joint inflammation, leading to functional impairments. Current treatments for autoimmune diseases often entail systemic side effects. Flavia et al. devised composite platforms comprising porous silicon (Psi) coated with KG-1 macrophage cell membranes (TCPSi@KG-1) for RA treatment [72]. Their results demonstrated that TCPSi@KG-1 significantly improved the stability of hydrophobic particles in both plasma and simulated synovial fluid. Cytocompatibility tests across different cell lines, including those from target organs, blood vessels, kidneys, and livers, revealed viability for up to 48 h, even at higher concentrations. Moreover, investigations into the immunological profile in KG-1 macrophages indicated that the nanoplatforms decreased immunostimulatory potential and prevented immune system activation.

##### Tumor Cell Membrane

Tumor cell membranes (TCMs) harvested from tumor cells are pivotal components in the development of innovative therapeutic vaccines, and have been utilized to trigger T cell immune response [73]. The use of TCMs in vaccine formulations is a promising approach that leverages the natural tumor-mimicking characteristics of cancer cells to prime and potentiate the immune system against cancers. Specifically, these membranes are naturally enriched with tumor-associated antigens, which can directly train the immune system to recognize and attack cancerous cells, boosting the anti-tumor response and fostering long-term immune memory [74]. The intrinsic properties of TCMs, including their capacity to mimic tumor cells, enable them to effectively target homologous tumor cells, offering a strategic advantage in cancer vaccine therapy [75]. Furthermore, these CNPs can be further functionalized with immune adjuvants to increase their efficacy, thus providing both targeted delivery and enhanced immunostimulation [76]. These advancements will not only improve the efficacy of cancer vaccines but also reduce side effects by minimizing off-target immune responses.

For example, Liu et al. developed a cancer vaccine by encapsulating immune adjuvant-loaded nanoparticles with camelina-derived cell (CC) membranes modified with mannose [76]. The Toll-like receptor 7 agonist IMQ (R837) was loaded into PLGA NPs (NP-R). Subsequently, NP-Rs were coated with mannose-modified CC membranes (NP-R@M-M), where surface proteins served as tumor antigens. The resultant NP-R@M-M exhibited enhanced uptake by DCs, promoting DC maturation and triggering a robust anti-tumor immune response in a B16F10 tumor-bearing mouse model. Similar strategies utilizing CpG oligodeoxynucleotide (CpG) as an immune adjuvant were also reported for combating cancer [77]. Similarly, Liu et al. integrated immunotherapy with starvation therapy using tumor cell-based CNPs, enhancing tumor targeting and therapy efficacy [78]. Santos et al. developed a tumor cell-based nanovaccine using an immunoadjuvant porous silicon (PSi) NP as the core, for achieving enhanced cancer immunotherapy [74]. This nanovaccine was biocompatible over a wide range of concentrations and could induce the expression of costimulatory signals, both in immortal cell lines and peripheral blood monocytes, as well as the secretion of pro-inflammatory cytokines such as IFN-γ.

Direct stimulation of immune responses through engineered CNPs represents another pivotal strategy in cancer vaccine design. Zhang et al. crafted nanoparticles coated with TCMs and engineered to express co-stimulatory markers, enabling direct T cell stimulation without professional APCs, thus offering a novel approach for multi-antigenic cancer immunotherapies [79] (Figure 5a). Additionally, Zhang et al. presented cellular nanodiscs crafted from TCMs and fortified with a lipid-based adjuvant for anti-tumor vaccination [80]. These nanodiscs, characterized by their diminutive size and disc-shaped morphology, were readily internalized by APCs and efficiently trafficked to the LNs. Due to their highly immunostimulatory properties, the nanodisc vaccine robustly activated the immune system and fostered tumor-specific immunity (Figure 5b). In a murine colorectal cancer model, significant control over tumor growth was observed in both preventive and therapeutic scenarios, particularly when combined with checkpoint blockade therapies.

Personalized cancer vaccine strategies utilize patient-specific TCMs to enhance vaccine specificity and efficacy. Mei et al. demonstrated the potential of photothermal vaccines created from surgically removed TCMs, which significantly enhanced DC activation and subsequent T cell responses, showing promise in preventing tumor relapse and metastasis [81]. Wei et al. developed a CNP platform based on photothermal nanoparticle-loaded tumor cells, which could be rationally applied to achieve on-demand boost of anti-tumor immune responses to inhibit tumor growth [82]. During the fabrication process, mild photothermal heating by near-infrared (NIR) laser irradiation induced the nanoparticle-bearing tumor cells to express heat shock proteins as endogenous adjuvants. After vaccination, non-invasive NIR laser irradiation further induced mild hyperthermia at the vaccination site, which promoted recruitment, activation, and antigen presentation by DCs. In addition, genetic engineering of tumor cell membranes to create enhanced vaccine platforms also plays a significant role in cancer immunotherapy. Huang et al. utilized antibody-anchored membrane technology to develop nanovaccines that effectively activate DCs and subsequent adaptive immune responses, showing enhanced anti-tumor efficacy in various tumor models [83].

#### 3.2.2. Bacterial Membrane-Coated Nanoparticles

In pursuit of innovative cancer therapies, bacterial CNPs have emerged as a pivotal technology in the development of therapeutic vaccines. Leveraging the unique immunostimulatory properties of bacterial outer membranes, these CNPs offer a promising strategy for enhancing the efficacy of cancer immunotherapy. The incorporation of pathogen-associated molecular patterns found on bacterial membranes can effectively stimulate the immune system by engaging with pattern recognition receptors on innate immune cells. This interaction is crucial for transforming immunologically “cold” tumors into “hot” ones, thus potentiating the body’s anti-tumor response.

In recent years, innovations in cancer immunotherapy have increasingly leveraged bacterial membrane components and nanotechnology to enhance the effectiveness of vaccines. These methods utilize bacterial membrane materials to advance vaccine delivery systems and improve immunological responses against cancer. For example, a multi-functional bacterial membrane-coated nanoparticle incorporating an immune activating PC7A/CpG polyplex core was combined with radiation therapy, and the design captured cancer neoantigens, enhancing DC uptake and T cell response in melanoma and neuroblastoma models, thus demonstrating an innovative approach to in situ cancer vaccines [84]. Hybrid vectors, which combine bacterial outer membrane vesicles (OMVs) with autologous TCMs, intensify the innate immune response and specifically target tumor heterogeneity, effectively reducing lung metastasis and highlighting the benefits of personalized immunotherapy [85]. Ping et al. coated OMVs on drug-loaded polymeric micelles to create an immune response [86]. Whereas the OMVs could activate the host immune response for cancer immunotherapy, the loaded chemotherapeutic agents in polymeric micelles exerted both chemotherapeutic and immunomodulatory roles to sensitize cancer cells to cytotoxic T cells and to kill cancer cells directly (Figure 6a). Nie et al. developed bacteria-derived OMVs as an mRNA delivery platform, by genetically engineering with surface decoration with an RNA binding protein, L7Ae, and a lysosomal escape protein, listeriolysin O (OMV-LL) [87]. OMV-LL could rapidly adsorb box C/D sequence-labelled mRNA antigens through L7Ae binding (OMV-LL-mRNA) and deliver them into DCs, followed by the cross-presentation via listeriolysin O-mediated endosomal escape. OMV-LL-mRNA significantly inhibited melanoma progression and elicited 37.5% complete regression in a colon cancer model (Figure 6b). The researchers from Nie’s group also constructed hybrid membrane-based vectors through fusion of the bacterial cytoplasmic membrane and the primary TCMs from surgically removed tumor tissues, possessing unique advantages as personalized cancer vaccines when neoantigens are not readily available [88]. In addition, the employment of bacterial cytoplasmic membranes in constructing antigen and adjuvant co-delivery nanoparticle vaccines was also proven to effectively induce tumor regression and extend survival in multiple cancer models by enhancing dendritic cell maturation and subsequent T cell activation [89]. 

**Table 1 vaccines-12-00603-t001:** Lipid-based vectors for advanced therapeutic vaccines.

Materials	Types	Role in Delivery	Applications	Ref.
Noncationic thiourea lipid	LNPs	Spleen targeting	mRNA delivery	[18]
2-(hexamethyleneimino) ethyl methacrylate (C7A-MA)	LNPs	Enhances endosomal escape	mRNA vaccine	[24,25]
DOPC	LNPs	Facilitates the mRNA transfection	Gene editing	[30]
PEG-DSPE	LNPs	Facilitates mRNA transfection	mRNA delivery	[31]
93-O17S-F	LNPs	Enhances cross-presentation and STING activation	In situ cancer vaccination	[34]
DSPC	LNPs	Secondary lymphatic organ targeting	mRNA vaccine	[35]
Phosphatidylserine (PS)	LNPs	Secondary lymphatic organ targeting	mRNA delivery	[20]
Photoisomerable Azo-based lipidoid	LNPs	Enhances endosomal escape	mRNA delivery	[23]
3β [L-histidinamide-carbamoyl] cholesterol (Hchol) lipid	LNPs	Enhances endosomal escape	mRNA delivery, potent antibody induction for SARS-CoV-2	[37]
Adjuvant lipidoid	LNPs	Enhances the adjuvanticity of the vaccine	mRNA vaccines	[38]
Non-nucleotide STING agonist-derived amino lipids (SALs)	LNPs	STING activation	mRNA vaccine for SARS-CoV-2	[39]
Ionizable lipid	LNPs	Immunosuppressive	Autoimmune conditions	[40]
Ganglioside	Liposomes	Enhances cytokine production and T cell activation	Cancer vaccine	[50]
Hen egg lysozyme (HEL)	Liposomes	B cell activation	Develop viral mimics for vaccine research	[51]
Poly(ethylenimine)	Liposomes	Enhances immune response	Antibacterial	[52]
Cobalt-porphyrin-phospholipid (CoPoP)	Liposomes	Enhancing cellular immune responses	Cancer vaccine	[53]
HMME and R837	Liposomes	Tumor-targeted delivery	Clinical cancer therapy	[55]
Influenza hemagglutinin (HA) immunogens	Liposomes	Enhances immunogenicity	Influenza vaccines	[56]
PEI and lipopeptide	Liposomes	Enhances mucosal and systemic immunity	Vaccines against Group A Streptococcus (GAS) infections	[58]
PDT sensitizers	Lipid-based hybrid nanoparticles	Enhances stability and endosomal escape	Tumor vaccine	[24,25]
PLGA polymeric core, lipid shell, DOX, Ce6	Lipid-based hybrid nanoparticles	Enhances immunogenic cell death	In situ tumor vaccination	[57]
LPH	Lipid-based hybrid nanoparticles	Enhances DC recruitment and activation, reduces immunosuppressive signals	Tumor vaccine	[59]
IMQ, MPLA, and OVA	Lipid-based hybrid nanoparticles	Enhances DC maturation	Tumor immunotherapy	[61]
Iron oxide core	Lipid-based hybrid nanoparticles	Lymph node targeting	Cancer immunotherapy	[62]
Radiolabeled iron oxide cores (Ga^+^)	Lipid-based hybrid nanoparticles	Lymph node targeting	Cancer immunotherapy	[63]
Nd/Yb/Er core, maleimide-functionalized lipids	Lipid-based hybrid nanoparticles	Trigger tumor ICD	In situ vaccination	[64]
Gold NPs	Lipid-based hybrid nanoparticles	Trigger tumor ICD	Cancer immunotherapy	[65]
SiPCCl_2_-hybridized mesoporous silica with Fe(III)-captopril	Immune-cell-member coated nanoparticles	Target H22 tumors, trigger ICD	Cancer immunotherapy	[68]
YK009 lipid	Immune-cell-member coated nanoparticles	Lymphatic targeting, enhance endosomal escape	SARS-CoV-2 immunization	[69]
DC-cancer fused membranes	Immune-cell-member coated nanoparticles	Targets homologous tumors and lymphoid	Cancer immunotherapy	[70]
Porous silicon (Psi) coated with KG-1 macrophage cell membranes (TCPSi@KG-1)	Immune-cell-member coated nanoparticles	Decreases immunogenicity	Treatment of rheumatoid arthritis (RA)	[72]
PLGA NPs loaded with IMQ (R837)	Tumor-cell-member coated nanoparticles	Triggers anti-tumor immune response	Cancer immunotherapy	[76]
CpG oligodeoxynucleotide	Tumor-cell-member coated nanoparticles	Enhances tumor targeting	Cancer immunotherapy	[77]
Immunoadjuvant porous silicon (PSi) NPs	Tumor-cell-member coated nanoparticles	Enhances immune response	Cancer immunotherapy	[74]
NPs coated with TCMs expressing co-stimulatory markers	Tumor-cell-member coated nanoparticles	Robust immune activation	Cancer vaccination	[79]
Black phosphorus quantum dot nanovesicles (BPQD-CCNVs) coated with surgically removed tumor cell membrane	Tumor-cell-member coated nanoparticles	Enhances DC activation, improved T cell responses	Cancer immunotherapy	[81]
Photothermal NP-loaded tumor cells	Tumor-cell-member coated nanoparticles	Robust immune activation	Cancer immunotherapy	[82]
Agonistic-antibody-boosted tumor cell	Tumor-cell-member coated nanoparticles	Enhances DC activation and adaptive immune responses	Cancer vaccination	[83]
OMV-coated drug-loaded polymeric micelles	Bacterial membrane-coated nanoparticles	Activation of host immune response	Cancer immunotherapy	[86]
Bacteria-derived OMVs (OMV-LL)	Bacterial membrane-coated nanoparticles	Enhances endosomal escape	Cancer immunotherapy	[87]

## 4. Immunogenicity and Disadvantages of Lipid-Based Vectors

In vaccine therapy, vaccines usually need to maintain strong immunogenicity to efficiently elicit antigen-specific immune responses. However, the immune system is very complex and the carriers of vaccines are required to show low immunogenicity. This is because the strong immunogenicity of the carriers will lead to an increase in the body’s defense capability against the carriers, making multiple-dose administration a challenge.

As summarized in Section 3, PEG-conjugated lipids are one of the most common components of lipid-based vectors. While free PEG typically exhibits low immunogenicity, it has been acknowledged as a polyvalent hapten capable of acquiring immunogenic properties when incorporated into nanoparticles [90]. This can lead to the induction of anti-PEG antibodies (mainly containing IgG and IgM against PEG) [91]. The anti-PEG antibodies have the potential to form antigen–antibody complexes with newly administered lipid-based vectors. Subsequent clearance of these complexes by mononuclear phagocytic cells may result in reduced therapeutic outcomes for lipid-based vectors [92]. Furthermore, antigen–antibody complexes may also induce severe hypersensitivity reactions. Therefore, strategies to reduce the immunogenicity of lipid-based vectors are essential. There are different ways to regulate the vectors’ immunogenicity, including (a) controlling the composition and characteristics of lipid-based vectors, and (b) changing the injection routes (include i.v., i.m., i.d., subcutaneous (s.c.), and intranasal (i.n.) injection) [93]. The appropriate administration route must be determined based on an understanding of the anatomy of the inoculation site and the induced immune responses. Recently, Yu et al. reported a novel polymer, poly(ethyl ethylene phosphate) (PEEP), with excellent degradability and stealth effects, which was employed as an alternative to PEG to overcome the “PEG dilemma” [94]. They found that PEEPylated lipid-based vectors exhibited lower immunogenicity and generated negligible anti-PEEP antibodies (IgM and IgG) after repeated treatments. Jiang et al. found that using zwitterionic poly(N-(3-acrylamidopropyl)carboxybetaine) (PCB) to replace the PEG also showed great potential for reducing immunogenicity [95]. One of the important obstacles currently hindering the development of lipid-based vectors is their high immunogenicity. Exploring strategies to reduce the immunogenicity of vectors may promote the optimization of related premarket guidelines and clinical protocols.

Lipid-based vectors also come with some disadvantages: (a) Limited cargo capacity: Lipid-based vectors have a relatively small cargo capacity compared to viral vectors. This limitation may restrict their use for delivering larger cargoes or multiple cargoes simultaneously. (b) Low delivery efficiency: Especially for gene delivery, gene expression mediated by lipid-based vectors is often transient. This means that the therapeutic effect might diminish over time, requiring repeated administrations for sustained benefit. (c) Variable delivery efficiency: The efficiency of lipid-based vectors can vary depending on factors such as cell type, lipid composition, and formulation techniques. Achieving consistent and high delivery efficiency across different cell types can be challenging. (d) Manufacturing complexity: The production of lipid-based vectors can be complex and require specialized equipment and expertise. Achieving consistency and scalability in manufacturing can pose challenges, particularly for clinical-grade vectors. (e) Storage and stability: Lipid-based vectors can be sensitive to storage conditions and may have limited stability. Ensuring proper storage and handling is crucial for maintaining their efficacy. (f) Off-target effects: Lipid-based vectors may exhibit off-target effects. This can potentially result in unintended physiological effects or adverse reactions. Thorough preclinical and clinical evaluation of lipid-based vectors is also essential, to assess their safety profile and potential risks in specific applications. (g) Risk of causing autoimmune diseases. Using immune cell membranes to construct vectors for therapeutic purposes is a promising area of research. However, concerns about potential immune reactions, including autoimmune diseases, must be carefully considered. Autoimmune diseases occur when the immune system mistakenly attacks healthy cells and tissues in the body. When using immune cell membranes for therapeutic vectors, there is a risk that the immune system could recognize these vectors as foreign and mount an immune response against them. This immune response could potentially lead to inflammation and tissue damage, similarly to what occurs in autoimmune diseases. Despite these disadvantages, lipid-based vectors remain valuable tools. Ongoing research aims to address some of these limitations through improved vector design and formulation techniques.

## 5. Summary and Outlook

The versatility of lipid-based vectors offers researchers multiple benefits when developing advanced therapeutic vaccines. Specifically, lipid-based vectors provide advantages including biocompatibility, biodegradability, safety, and scalability, rendering them an optimal choice for vaccine delivery systems. This review first discussed the recent strategies for delivering vaccines via lipid-based vectors to the lymphoid organs (Section 2). Significant progress has been made in the field of lymphoid organ targeting; however, it is crucial to recognize that the vectors may still accumulate in other organs, especially the liver. Continued refinement in their targeting mechanisms is essential for the safety improvement and eventual clinical translation of these advances, and this is a research topic that should be focused on in the next step. Designing lipid-based vectors that effectively facilitate endosome escape is also critical for vaccine therapy. Lipid-based vectors utilize strategies like the “proton sponge effect,” pH-responsive charge switching, and fusogenic lipids for endosomal escape during vaccine delivery (Section 2). However, it is worth noting that excessive endosomal and/or lysosomal disruption increase the risk of digestive enzyme leakage from endo-lysosomes. Future development directions should focus on the following two key points: (a) enhancing the capability of lipid-based vectors to escape to the cytoplasm from early endosomes with negligible digestive enzymes; (b) enabling lipid-based vectors to bypass the endocytosis pathway to enter cells. This review then discussed various synthetic lipid-based vectors, such as LNPs (Section 3.1.1), liposomes (Section 3.1.2), and lipid-based hybrid NPs (Section 3.1.3), which have been used to induce and strengthen the antigen-specific immune responses in preclinical experimental models. A variety of natural lipid-based vectors prepared using cell and/or bacterial membranes and their applications were also introduced (Section 3.2.1 and Section 3.2.2). As vaccine delivery systems, natural lipid-based vectors have unique advantages in acting as adjuvants for vaccine therapy compared with synthetic vectors. Finally, we proposed that, in order to promote the clinical translation of lipid-based vectors, it is imperative to address their immunogenicity (Section 4). Other issues of concern are maintaining batch-to-batch consistency of lipid-based vectors during fabrication and their scale-up to production at industrial levels, especially for natural lipid formulations.

Despite many successes in combating infection, successful application of current vaccine technologies toward cancer therapy has been an elusive goal. This is because tumor-derived resistance mechanisms develop during cancer progression, often inducing immune evasion. Additionally, compromised immune systems in cancer patients, resulting from factors like immune cell depletion, age, or immunotherapy side effects, can hinder the immune response stimulated by vaccines. Therefore, in addition to innovations in lipid-based material design, the next generation of vaccines will likely benefit from combination adjuvant approaches, which can target multiple branches of the immune response simultaneously. The optimal activation of immunity could be achieved by harnessing the immunostimulatory properties of various adjuvants. When these adjuvants are integrated into lipid-based vaccine, they show the potential to further amplify both the quality and quantity of the immune response against targeted antigens.

## Figures and Tables

**Figure 1 vaccines-12-00603-f001:**
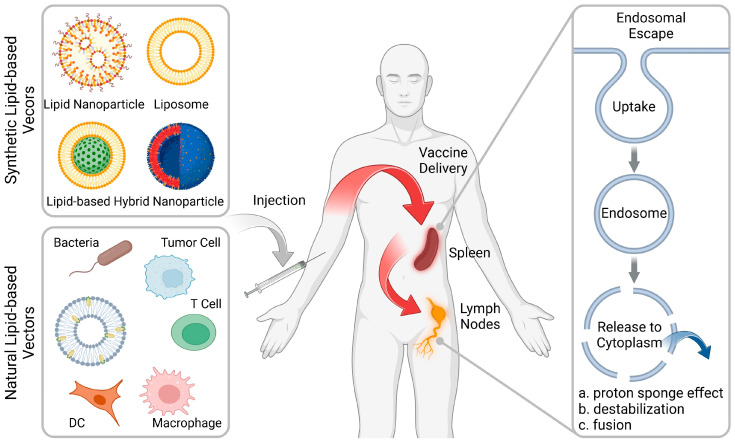
Rational design of lipid-based vectors for advanced therapeutic vaccines. Lipid-based vectors (including synthetic and natural formulations) deliver the vaccines into the lymphoid organs, including the spleen and lymph nodes (LNs), followed by facilitating the endosomal escape for robust immune activation. (Figure created with BioRender.com).

**Figure 4 vaccines-12-00603-f004:**
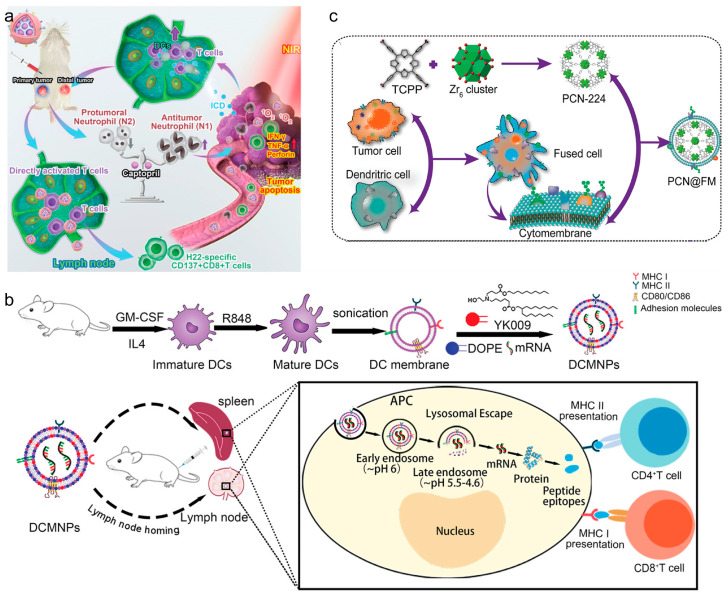
(**a**) Remodeling tumor-associated neutrophils to enhance dendritic-cell-based HCC neoantigen nano-vaccine efficiency. Reproduced from ref. [68]. Copyright 2022 Wiley-VCH. (**b**) Dendritic-cell-mimicking nanoparticles promote mRNA delivery to lymphoid organs. Reproduced from ref. [69]. Copyright 2023 Wiley-VCH. (**c**) Expandable immunotherapeutic nanoplatforms engineered from cytomembranes of hybrid cells derived from cancer and dendritic cells. Reproduced from ref. [70]. Copyright 2019 Wiley-VCH.

**Figure 5 vaccines-12-00603-f005:**
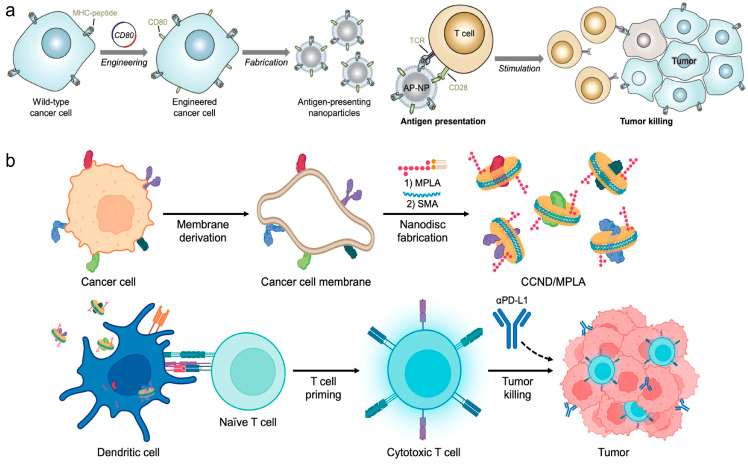
(**a**) Engineered cell-membrane-coated nanoparticles directly present tumor antigens to promote anticancer immunity. Reproduced from ref. [79]. Copyright 2020 Wiley-VCH. (**b**) Cancer cell membrane nanodiscs for antitumor vaccination. Reproduced from ref. [80]. Copyright 2023 American Chemical Society.

**Figure 6 vaccines-12-00603-f006:**
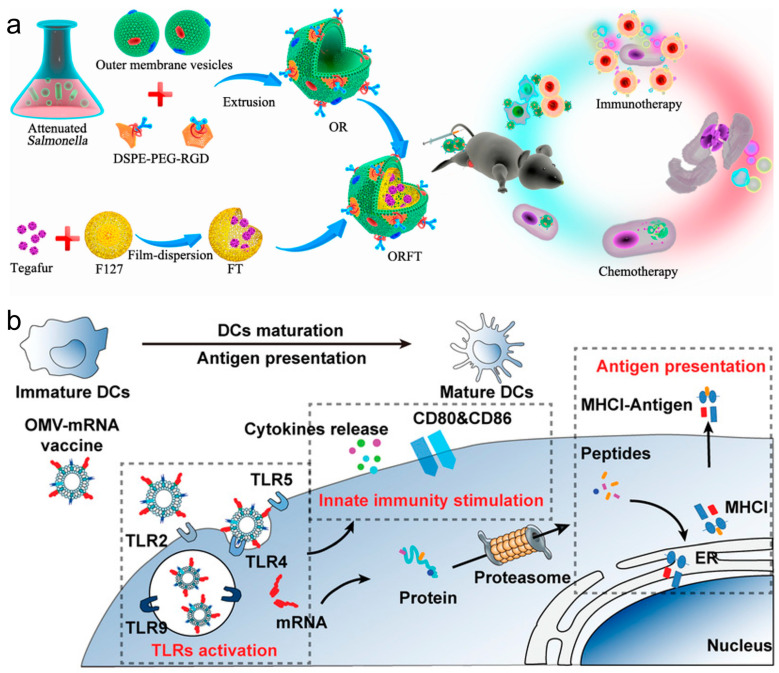
(**a**) Bioengineering bacterial vesicle-coated polymeric nanomedicine for enhanced cancer immunotherapy and metastasis prevention. Reproduced from ref. [86]. Copyright 2020 American Chemical Society. (**b**) Rapid surface display of mRNA antigens by bacteria-derived outer membrane vesicles for a personalized tumor vaccine. Reproduced from ref. [87]. Copyright 2022 Wiley-VCH.

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
