# Peer review of "Rational Design of Lipid-Based Vectors for Advanced Therapeutic Vaccines"

_vaccines, 2024, doi:10.3390/vaccines12060603_

Round 1

Reviewer 1 Report

Comments and Suggestions for Authors

This review is very complet; however I strongly suggest to include the possible disadvantages of theses components lipids base vector into the individual tht receive these kind of vectors or even RNA vaccines  alone. I mean these vectors could accelerate trombos formation or induce any kind of autoimmune reaction? Explain deeply, using text and figures. 

Author Response

Thanks for your suggestions. We have included possible disadvantages of lipid-based vectors in the revised manuscript, including the risk of autoimmune diseases caused by immune cell membrane-based vectors. We have added the following discussion to the Section 4.

“Lipid-based vectors also come with some disadvantages: a) Limited cargo capacity: Lipid-based vectors have a relatively small cargo capacity compared to viral vectors. This limitation may restrict their use for delivering larger cargoes or multiple cargoes simultaneously. b) Low delivery efficiency: Especially for gene delivery, gene expression mediated by lipid-based vectors is often transient. This means that the therapeutic effect might diminish over time, requiring repeated administrations for sustained benefits. c) Variable delivery efficiency: The efficiency of lipid-based vectors can vary depending on factors such as cell type, lipid composition, and formulation techniques. Achieving consistent and high delivery efficiency across different cell types can be challenging. d) Manufacturing complexity: The production of lipid-based vectors can be complex and require specialized equipment and expertise. Achieving consistency and scalability in manufacturing can pose challenges, particularly for clinical-grade vectors. e) Storage and stability: Lipid-based vectors can be sensitive to storage conditions and may have limited stability. Ensuring proper storage and handling is crucial to maintaining their efficacy. f) Off-target effects: Lipid-based vectors may exhibit off-target effects. This can potentially result in unintended physiological effects or adverse reactions. Thorough preclinical and clinical evaluation of lipid-based vectors is also essential to assess their safety profile and potential risks in specific applications. g) Risk of causing autoimmune diseases. Using immune cell membranes to construct vectors for therapeutic purpose is a promising area of research. However, concerns about potential immune reactions, including autoimmune diseases, must be carefully considered. Autoimmune diseases occur when the immune system mistakenly attacks healthy cells and tissues in the body. When using immune cell membranes for therapeutic vectors, there is a risk that the immune system could recognize these vectors as foreign and mount an immune response against them. This immune response could potentially lead to inflammation and tissue damage, similar to what occurs in autoimmune diseases. Despite these disadvantages, lipid-based vectors remain valuable tools. Ongoing research aims to address some of these limitations through improved vector design and formulation techniques.”

Reviewer 2 Report

Comments and Suggestions for Authors

This review paper fully discusses the advantages and disadvantages of the rational design of lipid-based vectors for the vaccine delivery. The authors summarized six reasons to explain why lipid delivery is an optimal choice to deliver antigens and mRNAs from degradation and aggregation. This review focused on two out of six, including effective delivery of vaccines to primary lymphoid organ draining to secondary lymphoid tissues and endosome escape. In addition, the review also discusses the advantages, applications, and limitations of synthetic lipid nanoparticles and natural system coated nanovesicles. This system includes diseased cell membrane (tumor) and non-diseased cell membrane (immune system), bacterial membrane system. It highlights the potential promising applications in cancer immunotherapy.

In addition to discuss the tumor, I think that it is also worth to discuss the inflammation and autoimmune disease membranes as well since this is another main part that is associated with immune system.

Author Response

(The authors gave the same response as above.)

Reviewer 3 Report

Comments and Suggestions for Authors

A review article by Ma et al describes rational design of lipid-based vectors for therapeutic vaccine.  It is well written article and with an up-to-date review which contribute substantive information that will add to vaccine therapeutic strategy.  The review can be enhanced with minor modifications listed below.

1.     Figure 1 legend needs additional description adding information for the cartoons depicting the synthetic and natural lipid-based vectors.

2.     Section 3:  For easy read to the general audience and better understanding, a Table is necessary in section 3 listing the different lipid vectors described with their structure (or/and citing reference in the table next to the names) as well as phrases of findings or application. Such a Table would enhance the quality of this Review and easy read.

Author Response

We have carefully checked and revised our manuscript according to the editor’s requirements and the reviewers' suggestions and comments. The changes have been highlighted in the revised manuscript.

Reviewer 4 Report

Comments and Suggestions for Authors

Since the success of COVID-19 vaccines, lipid-nanoparticle-based strategies to deliver the antigens have gained much attention. Several platforms have been exploring different LNPs for targeted delivery without off-target effects. Yufei Ma et al. have submitted a review manuscript entitled “Rational Design of Lipid-Based Vectors for Advanced Therapeutic Vaccines.” The authors reasonably comprehended the different lipid nanoparticle-based strategies and discussed their engineering for targeted delivery. Although the authors made substantial contributions to discussing novel strategies in LNPs, they have failed in some respects, which limits the manuscript publication in its current form.

Before making a decision about the manuscript, consider the following factors:

In the abstract, the authors stated, “It also discusses the importance of rational formulation design and structure-activity relationships..."  however, I have not seen any SAR-based discussion in the text or the table format.

Figure 1 failed to clearly represent the endosomal escape. Please represent the endosomal escape mechanism at the molecular level in the illustration.

Line 185: Which delivery system?

I believe this statement has already been made in lines 121–123.

Figure 2 and other figures look like they were reproduced from several publications. However, I don't see the copyrighted statement or permission-granted statements. Please get permission from all publications to use them as such or with minor modifications as well.

Importantly, the authors have not discussed any lipids or lipoids, their role in delivery, endosomal escape, or others. I would recommend creating a table with novel lipids or lipoids and their role in delivery, which adds novelty to the manuscript.

Author Response

(The authors gave the same response as above.)

Round 2

Reviewer 4 Report

Comments and Suggestions for Authors

The manuscript was amended based on the reviewers' suggestions. The manuscript is ready for publication. However, before publication, I recommend that the authors incorporate the copyright statement in the figures' legends and cite the relevant references.